# ONE-SHOT AND FEW-SHOT LEARNING OF WORD EMBEDDINGS

## ABSTRACT

Standard deep learning systems require thousands or millions of examples to learn a concept, and cannot integrate new concepts easily. By contrast, humans have an incredible ability to do one-shot or few-shot learning. For instance, from just hearing a word used in a sentence, humans can infer a great deal about it, by leveraging what the syntax and semantics of the surrounding words tells us. Here, we draw inspiration from this to highlight a simple technique by which deep recurrent networks can similarly exploit their prior knowledge to learn a useful representation for a new word from little data. This could make natural language processing systems much more flexible, by allowing them to learn continually from the new words they encounter.

## 1 INTRODUCTION

Humans are often able to infer approximate meanings of new words from context. For example, consider the following stanza from the poem "Jabberwocky" by Lewis Carroll:

> *He took his vorpal sword in hand:*
> *Long time the manxome foe he sought*
> *So rested he by the Tumtum tree,*
> *And stood awhile in thought.*

Despite the fact that there are several nonsense words, we can follow the narrative of the poem and understand approximately what many of the words mean by how they relate other words. This a vital skill for interacting with the world – we constantly need to learn new words and ideas from context. Even beyond language, humans are often able adapt quickly to gracefully accomodate situations that differ radically from what they have seen before. Complementary learning systems theory (Kumaran et al., 2016) suggests that it is the interaction between a slow-learning system that learns structural features of the world (i.e. a deep-learning like system) and a fast-learning system (i.e. a memory-like system) that allows humans to adapt rapidly from few experiences.

By comparison, standard deep learning systems usually require much more data to learn a concept or task, and sometimes generalize poorly Lake et al. (2017). They can be trained to learn a concept in one-shot if this is their sole task (Vinyals et al., 2016, e.g.), but this limits the types of tasks that can be performed. Furthermore, these models typically discard this information after a single use. In order for deep learning systems to be adaptible, they will need to build on their prior knowledge to learn effectively from a few pieces of information. In other words, they will need to integrate learning experiences across different timescales, as complementary learning systems theory suggests that humans and other animals do. In this paper, we explore this broad issue in the specific context of creating a useful representation for a new word based on its context.

### 1.1 BACKGROUND

Continuous representations of words have proven to be very effective (Mikolov et al., 2013; Pennington et al., 2014, e.g.). These approaches represent words as vectors in a space, which are learned from large corpuses of text data. Using these vectors, deep learning systems have achieved success on tasks ranging from natural language translation (Wu et al., 2016, e.g.) to question answering (Santoro et al., 2017, e.g.).

However, these word vectors are typically trained on very large datasets, and there has been surprisingly little prior work on how to learn embeddings for new words once the system has been trained. Cotterell et al. (2016) proposed a method for incorporating morphological information into word embeddings that allows for limited generalization to new words (e.g. generalizing to unseen conjugations of a known verb). However, this is not a general system for learning representations for new words, and requires building in rather strong structural assumptions about the language and having appropriately labelled data to exploit them.

More recently Lazaridou et al. (2017) explored multi-modal word learning (similar to what children do when an adult points out a new object), and in the process suggested a simple heuristic for inferring a word vector from context: simply average together all the surrounding word vectors. This is sensible, since the surrounding words will likely occur in similar contexts. However, it ignores all syntactic information, by treating all the surrounding words identically, and it relies on the semantic information being linearly combinable between different word embeddings. Both of these factors will likely limit its performance.

Can we do better? A deep learning system which has been trained to perform a language task must have learned a great deal of semantic and syntactic structure which would be useful for inferring and representing the meaning of a new word. However, this knowledge is opaquely encoded in its weights. Is there a way we can exploit this knowledge when learning about a new word?

## 2 APPROACH

We suggest that we already have a way to update the representations of a network while accounting for its current knowledge and inferences – this is precisely what backpropagation was invented for! Of course, we cannot simply train the whole network to accomodate this new word, this would lead to catastrophic interference. However, Rumelhart & Todd (1993) showed that a simple network could be taught about a new input by freezing all its weights except for those connecting the new input to the first hidden layer, and optimizing these by gradient descent as usual. They showed that this resulted in the network making appropriate generalizations about the new input, and by design the training procedure does not interfere with the network's prior knowledge. They used this as a model for human concept learning (as have other authors, for example Rogers & McClelland (2004)).

We take inspiration from this work to guide our approach. To learn from one sentence (or a few) containing a new word, we freeze all the weights in the network except those representing the new word (in a complex NLP system, there may be more than one such set of weights, for example the model we evaluate has distinct input and output embeddings for each word). We then use stochastic gradient descent ($\eta = 0.01$) to update the weights for the new word using 100 epochs of training over the sentence(s) containing it.

Of course, there are a variety of possible initializations for the embeddings before optimizing. In this paper, we consider three possibilities:

1. Beginning with an embedding for a token that was placed in the softmax but never used during training (for the purpose of having a useful initialization for new words). This might help separate the new embedding from other embeddings.

2. Beginning with a vector of zeros.

3. Beginning with the centroid of the other words in the sentence, which Lazaridou et al. (2017) suggested was a useful estimate of an appropriate embedding.

We compare these to two baselines:

1. The centroid of the embeddings of the other words in the sentence Lazaridou et al. (2017).

2. Training the model with the 10 training sentences included in the corpus from the beginning (i.e. the "standard" deep-learning approach).

(A reader of an earlier draft of this paper noted that Herbelot & Baroni (2017) independently tried some similar strategies, but our results go farther in a variety of ways, see Appendix E.)

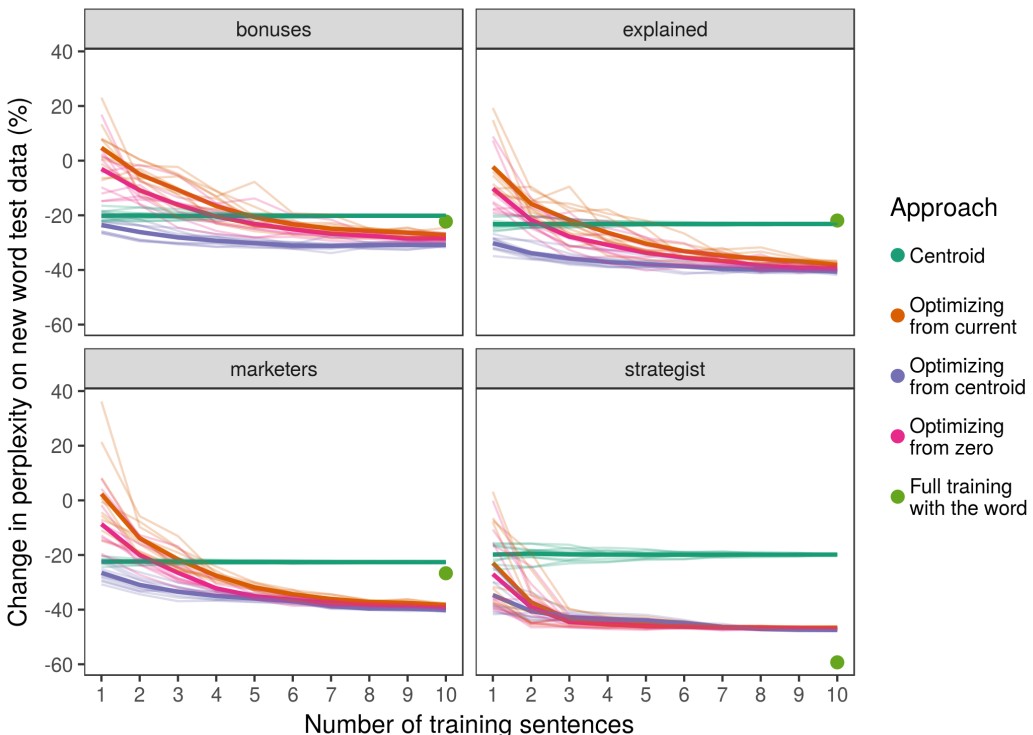

Figure 1: Percent change in perplexity on 10 test sentences containing new word, plotted vs. the number of training sentences, across four different words, comparing optimizing from three different starting points to centroid and training with the word baselines. Averages across 10 permutations are shown in the dark lines, the individual results are shown in light lines. (Note that the full training with the word was only run once, with a single permutation of all 10 training sentences.)

## 2.1 TASK, MODEL, AND APPROACH

The framework we have described for updating embeddings could be applied very generally, but for the sake of this paper we ground it in a simple task: predicting the next word of a sentence based on the previous words, on the Penn Treebank dataset (Marcus et al., 1993). Specifically, we will use the (large) model architecture and approach of Zaremba et al. (2014), see Appendix B.1 for details.

Of course, human language understanding is much more complex than just prediction, and grounding language in situations and goals is likely important for achieving deeper understanding of language (Gauthier & Mordatch, 2016). More recent work has begun to do this (Hermann et al., 2017, e.g), and it's likely that our approach would be more effective in settings like these. The ability of humans to make rich inferences about text stems from the richness of our knowledge. However, for simplicity, we have chosen to first demonstrate it on the prediction task.

In order to test our one-shot word-learning algorithm on the Penn Treebank, we chose a word which appeared only 20 times in the training set. We removed the sentences containing this word and then trained the model with the remaining sentences for 55 epochs using the learning rate decay strategy of Zaremba et al. (2014). Because the PTB dataset contains over 40,000 sentences, the 20 missing ones had essentially no effect on the networks overall performance. We then split the 20 sentences containing the new word into 10 train and 10 test, and trained on 1 - 10 of these in 10 different permutations (via a balanced Latin square (Campbell & Geller, 1980), which ensures that each sentence was used for one-shot learning once, and enforces diversity in the multi-shot examples). In other words, we performed 100 training runs for each word, 10 training runs each with a distinct single sentence, 10 with a distinct pair of sentences, etc.

## 3 RESULTS

We first evaluated our approach on the words "bonuses," "explained," "marketers," and "strategist," either initializing from the never-seen embedding that the network is optimized to never produce, the zero vector, or the centroid of the surrounding words, and compared to the baselines of just taking the centroid of the surrounding words and full training with the words, see Fig. 1. Optimizing from the centroid outperforms all other approaches for learning a new word for all datasets, including the centroid approach of Lazaridou et al. (2017), and even outperforms full training with the word[1] in three of the four cases (however, this likely comes with a tradeoff, see below). The optimizing approaches are strongly affected by embedding initialization with few training sentences (e.g. one-shot learning), but by 10 sentences they all perform quite similarly.

Of course, this learning might still cause interference with the networks prior knowledge. In order to evaluate this, we replicated these findings with four new words ("borrow," "cowboys", "immune," and "rice"), but also evaluated the change in perplexity on the PTB test corpus (see Appendix A, Fig. 5). Indeed, we found that while the centroid method does not substantially change the test perplexity on the corpus, the optimizing method causes increasingly more interference as more training sentences are provided, up to a 1% decline in the case of training on all 10 sentences.

This is not necessarily surprising, the base rate of occurrence of the new word in the training data is artificially inflated relative to its true base rate probability. This problem of learning from data which is locally highly correlated and biased has been solved in many recent domains by the use of replay buffers to interleave learning, especially in reinforcement learning (Mnih et al., 2015, e.g). The importance of replay is also highlighted in the complementary learning systems theory (Kumaran et al., 2016) that helped inspire this work. We therefore tested whether using a replay buffer while learning the new word would ameliorate this interference. Specifically, we sampled 100 negative sentences from the without-word corpus the network was pre-trained on, and interleaved these at random with the new word training sentences (the same negative samples were used every epoch).

Indeed, interleaving sentences without the word substantially reduced the interference caused by the new word, see Fig. 2. The maximum increase in perplexity was 0.06%. This interleaving did result in somewhat less improvement on the new-word test sentences, but this is probably simply because the test sentences over-represent the new word and the network was overfitting to this and predicting the new word much more than is warranted. The optimizing approach still reduces perplexity on new word dataset by up to 33% (about 10 percentage points better than the centroid approach).

### 3.1 WHERE IS THE MAGIC HAPPENING?

Because the model we are using has distinct input and output embeddings, we are able to evaluate their distinct contributions to learning about the new word. Specifically, we compared learning only the softmax weights and bias (output embedding), to learning only the input embedding, as well as to learning both, for one- and ten-shot learning[2]. See Fig. 3 for our results.

We found that the changes in the output embeddings were almost entirely responsible for the overall improvement. In one-shot learning, changing the input embedding alone causes almost no improvement, and changing both embeddings does not seem substantially different from changing just the output embedding. However, with ten training sentences the updated input embedding is producing some improvement, both alone and when trained together with the output embedding. Even in this case, however, the effect of the input embedding is still much smaller than the effect of the output embedding. From this evidence, it seems likely that the model is mostly improving in predicting the new word in context, rather than predicting context based on the new word.

This is sensible for several reasons. First, whatever the new word conveys about the context will already be partly conveyed by the other words in the context. Second, our training approach was

---

[1]It is interesting to note that only one of these words appears in the PTB test data ("cowboys"), and it only appears once. Why then does full training with the word result in lowered test perplexity on the PTB test data in three out of four cases? This may just be chance variation, of course, but in general we would expect learning about a word to be useful not just for the sake of learning about that word, but because each word is a small piece of the signal by which the network learns about language more generally.

[2]Note that these analyses were conducted before we incorporated the replay buffer, but we expect the general pattern of the results would not be altered by including replay.

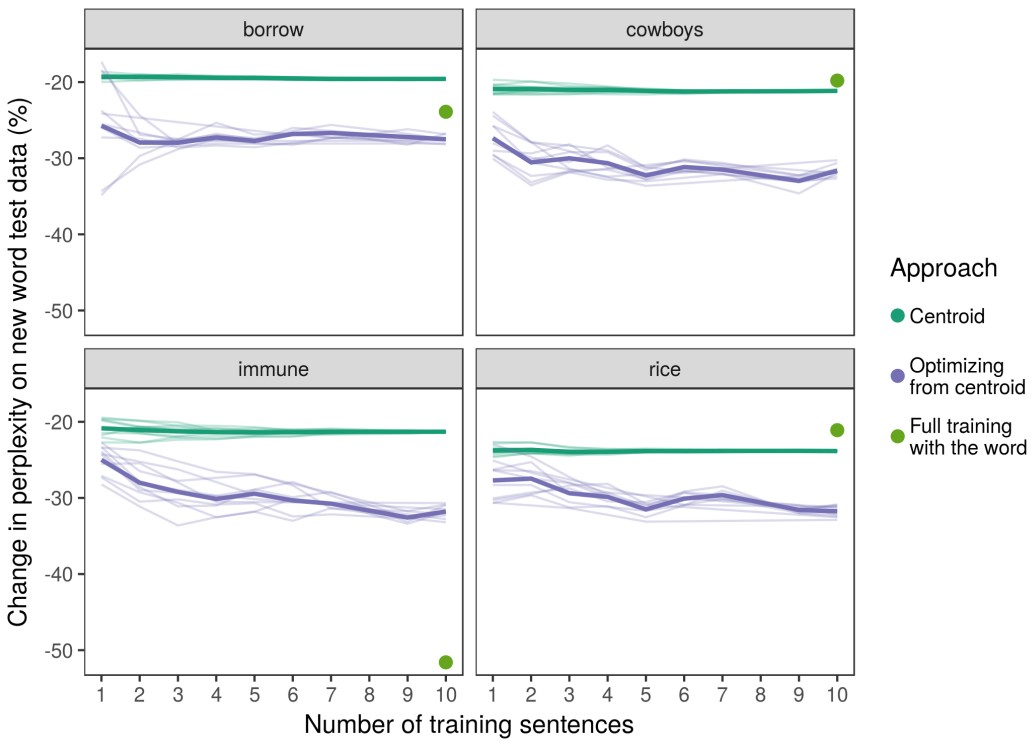

(a) Percent change in perplexity on 10 test sentences containing new word.

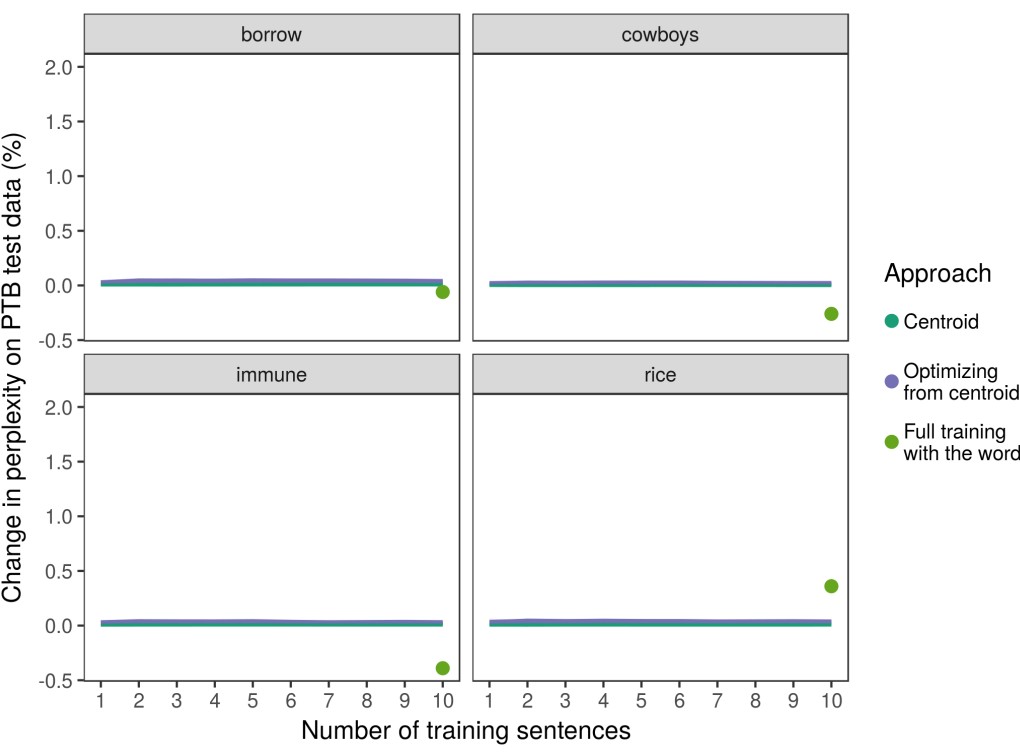

(b) Percent change in perplexity on full PTB test corpus.

Figure 2: Comparing full training with the word, centroid, and optimizing from the centroid approaches on both the new word dataset and the full test corpus (to assess interference), while using 100 negatively sampled sentences for replay. When using a replay buffer, learning new words does not interfere substantially with prior knowledge.

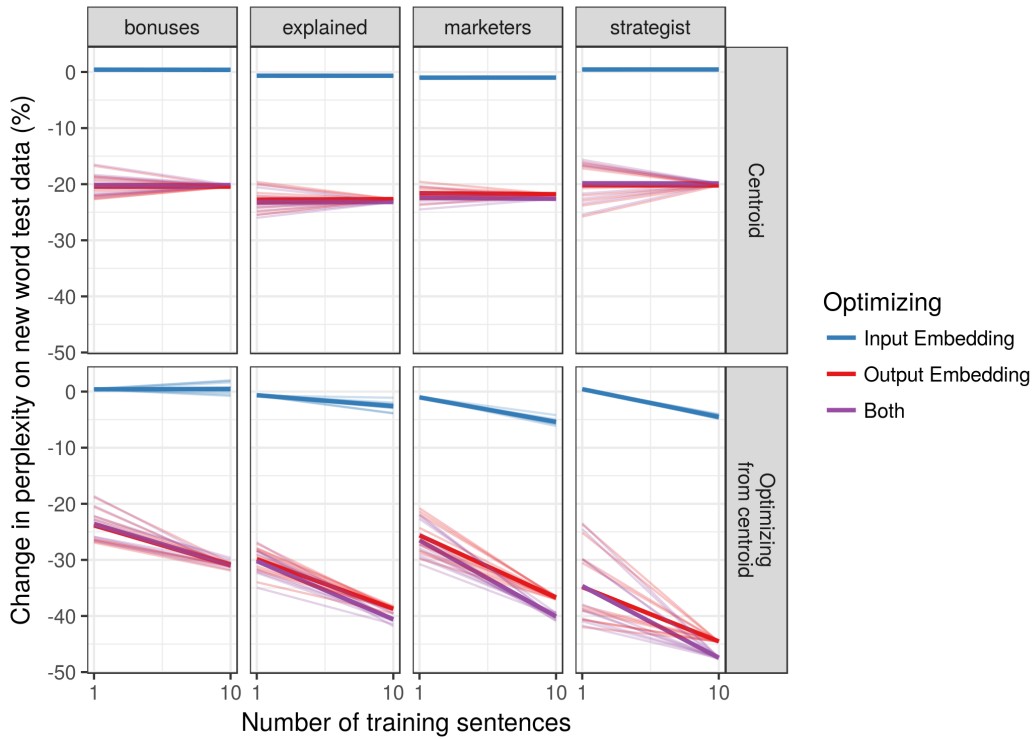

Figure 3: Comparing change in perplexity on the new word test set when optimizing the input embedding, output embedding, or both on either 1 or 10 sentences containing the new word. Light lines are 10 independent runs, dark lines are averages.

more unnatural than the situations in which a human might experience a new word, or even than the pretraining for the network, in that the sentences were presented without the context of surrounding sentences. This means that the model has less data to learn about how the new word predicts surrounding context, and less information about the context which predicts this word. This may also explain why full training with the word still produced better results in some cases than updating for it. Finally, efficiently passing information about the new word through the model from the input might require adjustments to the intermediate weights, which were frozen.

## 3.2 DIGGING DEEPER INTO MODEL PERFORMANCE

|  | New word is correct | Wrong but relevant | Wrong and irrelevant |
|---|---|---|---|
| Full training with the word | -9.21 | -12.75 | -15.13 |
| Centroid | -9.16 | -9.46 | -10.44 |
| Optimizing from centroid | -6.20 | -9.32 | -10.91 |

Table 1: Average log-probabilities of new word when: the word is the current target, the new word is not the current target but does appear in the current sentence, and the word doesn't appear in the sentence or context. (Analysis computed with 10 training sentences, patterns are similar but less severe with fewer sentences, see Appendix C.)

In order to dig more deeply into the effects of learning the new words embedding, we conducted more detailed analyses of the models predictions of the probability of the new word in different contexts. Specifically, we evaluated how well the word was predicted in three cases: when it was the actual target word, when it was not the current target but did appear in the sentence ("wrong but relevant") and when it was did not even appear in the sentence at all ("wrong and irrelevant"). This allowed us to investigate whether the model was learning something useful, or simply overfitting to the new data. We compared the average log-probability the model assigned to the new word in each of these cases

for the full training with the word baseline, the centroid approach to learning from 10 words, and our approach (with a replay buffer). The relevant cases were evaluated on our held-out test data for that word; the irrelevant case was evaluated on the first 10 sentences of the PTB test corpus (an article that does not contain any of the words we used). See Table 1 for our results.

The model fully trained with the word shows clear distinctions between the three conditions – the word is estimated to be about 10 times more likely in contexts where it appears than in irrelevant contexts, and is estimated to be about 25 times more likely again when it actually appears. However, the model severely underestimates the probability of the word when it does appear; the word would have a similar log probability under a uniform distribution over the whole vocabulary. The centroid method also has this issue, but in addition it does not even distinguish particularly well between contexts. The word is only estimated to be about 4 times more likely when it is the target than in completely irrelevant contexts.

By contrast, our approach results in a good distinction between contexts – the word is predicted to be about 5 times as likely in contexts where it appears compared to the irrelevant context, and about 25 times as likely again when the word actually appears. These relative probabilities are quite similar to those exhibited by the model fully trained with the word. In both respects, it appears superior to the centroid approach. When compared to the full training with the word, however, it appears that the base rate estimates of the prevalence of the word are inflated (which is sensible, even with 100 negative samples per positive sample in our training data the prevalence of the word is much higher than in the actual corpus). This explains the residual small increase in test perplexity on the dataset not containing the new word. It is possible that this could be ameliorated either by setting a prior on the bias for the new word (perhaps penalize the $\ell_2$ norm of the distance of the bias from the values for other rare words), or with a validation set, or just by using more negative samples during training. In any case, the optimized embeddings are capturing some of the important features of when the word does and does not appear, and are doing so more effectively than the centroid approach.

## 3.3 MORE WORDS

Up to this point, we have presented all the results in this paper broken down by word rather than as averages, because there were large word-by-word differences on almost every analysis, and we evaluated on relatively few new words. In order to establish the generality of our findings with a large sample of words, we ran an additional experiment that spans the space of words more broadly.

In this experiment, we selected 100 of the  150 words that appear exactly 20 times in the PTB train corpus (omitting the words we used in prior experiments, see Appendix B.1.3 for a complete list). Instead of training separate models without each word as we had previously, we trained a single model with **none** of these words included in the train set. We then tested our few-shot learning technique with a replay buffer (optimizing from centroid) and the centroid technique on these sentences, and compared to results obtained from full training with all words – a model trained with the entire train corpus, including the train sentences for each of the hundred words. (In all cases, the same 10 of the 20 sentences containing the new word were used in training, and the other 10 were used for testing.) Notice that the comparison to "full training with all words" is not as precise as our previous experiments – the model receives about 2.5 % more training data overall than any of the few-shot learning models, which means it will have more linguistic structure to learn the new words from, as well as the advantage of interleaving them. However, the comparisons between our technique and the centroid technique are still valid, and the comparison to the full training with all words gives a **worst-case** bound on how poorly one-shot methods will do compared to full training. With this in mind, see Fig. 4 (and Appendix A Fig. 6) for our results.

As before, optimizing from the centroid performed much better than simply using the centroid – on average it produced a 64% (11 percentage point) improvement over the centroid result, and on none of the 100 words did it perform worse than the centroid method. More quantitatively, the optimizing method performed significantly better (paired $t$-test, $t(99) = -20.5, p < 1 \cdot 10^{-16}$). Furthermore, despite the disadvantage of being exposed to less total data, the optimizing approach seems to do approximately as well as the full training approach on average – although the full training approach sometimes results in much larger improvements, the results did not significantly differ (paired $t$-test, $t(99) = 0.9, p = 0.39$). Across a wide variety of words, optimizing improves on the centroid approach, and performs comparably to full training.

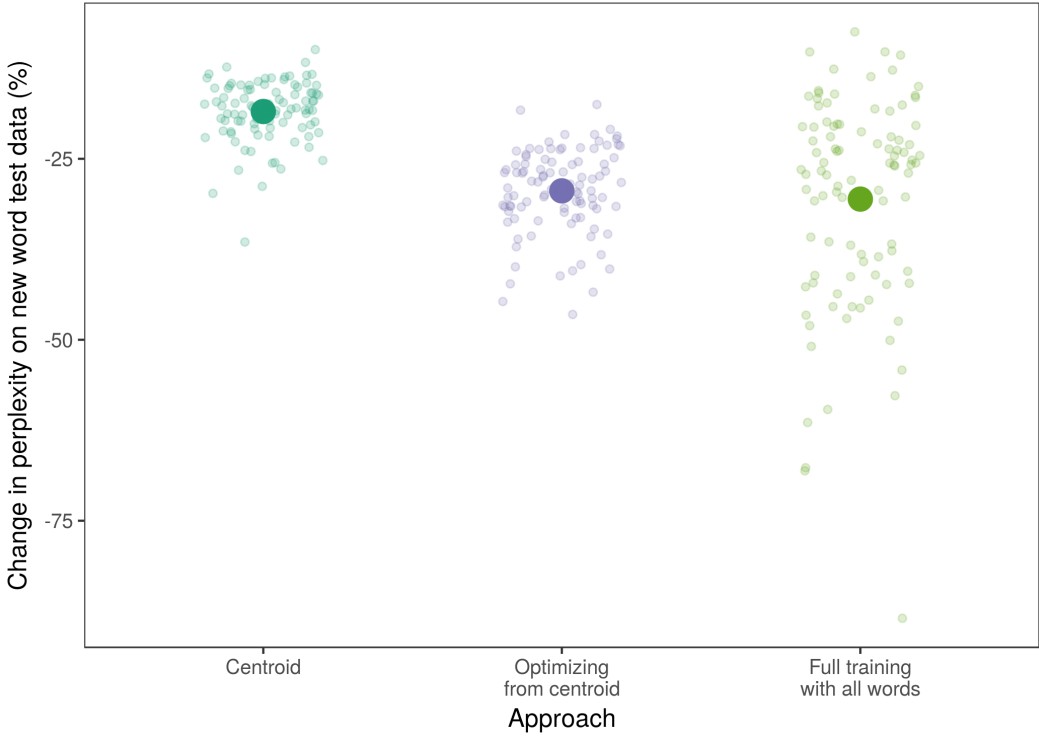

Figure 4: Percent change in perplexity on 100 new words from applying the centroid, optimizing from the centroid, and full training with all words. Ten sentences containing the new word were used in training and 10 were used in testing. Large solid dots indicate the change in the mean, smaller dots indicate the change for individual words.

## 4    DISCUSSION

Overall, using our technique of updating only the embedding vectors of a word while training on sentences containing it and negative sampled sentences from the networks past experience seems quite effective. It allows for substantial reductions in perplexity on text containing the new word, without greatly interfering with knowledge about other words. Furthermore, it seems to be capturing more useful structure about how the word is used in context than previous approaches, and performs close to as well as full training with the word. These results are exciting beyond their potential applications to natural language processing – this technique could easily be extended to adapting systems to other types of new experiences, for example a vision network for an RL agent could have a few new filters per layer added and trained to accomodate a new type of object.

Under what circumstances will this strategy fail? Complementary learning systems theory (Kumaran et al., 2016), from which we drew inspiration, suggests that information which is *schema-consistent* (i.e. fits in with the network's previous knowledge) can be integrated easily, whereas *schema-inconsistent* knowledge (i.e. knowledge that differs from the network's previous experience) will cause interference. Similar principles should apply here. Our approach should work for learning a new word on a topic which is already somewhat familiar, but would likely fail to learn from a new word in a context that is not well understood. For example, it would be difficult to learn a new German word from context if the model has only experienced English.

On the other hand, this perspective also provides promises. We expect that our technique would perform even better in a system that had a more sophisticated understanding of language, because it would have more prior knowledge from which to bootstrap understanding of new words. Thus it would be very interesting to apply our technique on more complicated tasks like question answering, such as Santoro et al. (2017), or in a grounded context, such as Hermann et al. (2017).

## 5 CONCLUSIONS

We have presented a technique for doing one- or few-shot learning of word embeddings from text data: freeze all the weights in the network except the embeddings for the new word, and then optimize these embeddings for the sentence, interleaving with negative examples from network's prior experience and stopping early. This results in substantial improvement of the ability to predict the word in context, with minimal impairment of prediction of other words. This technique could allow natural language processing systems to adapt more flexibly to a changing world, like humans do. More generally, it could serve as a model for how to integrate rapid adaptation into deep learning systems.

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

# A  Supplementary figures

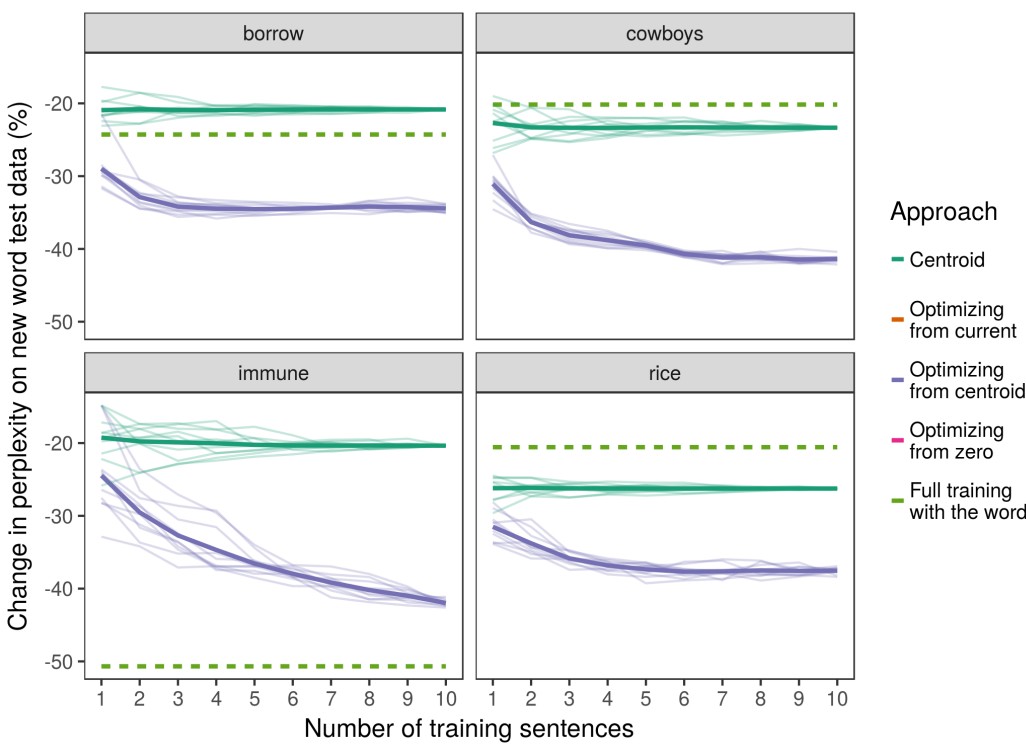

(a) Percent change in perplexity on 10 test sentences containing new word.

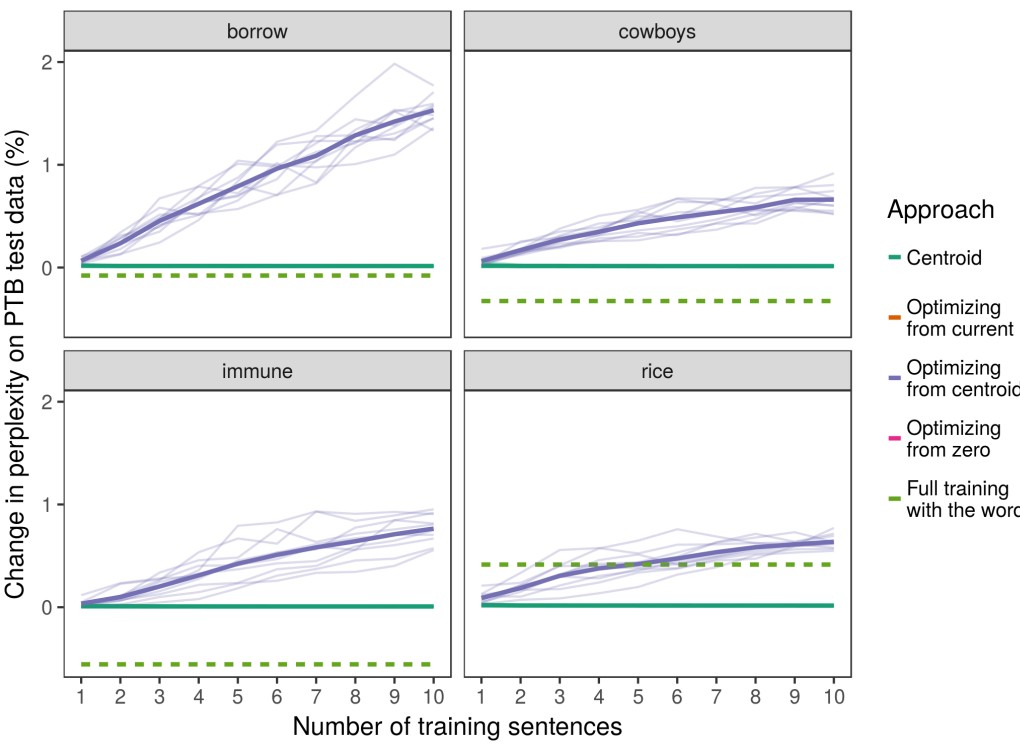

(b) Percent change in perplexity on full PTB test corpus.

Figure 5: Interference with prior knowledge caused by naïve use of our approach

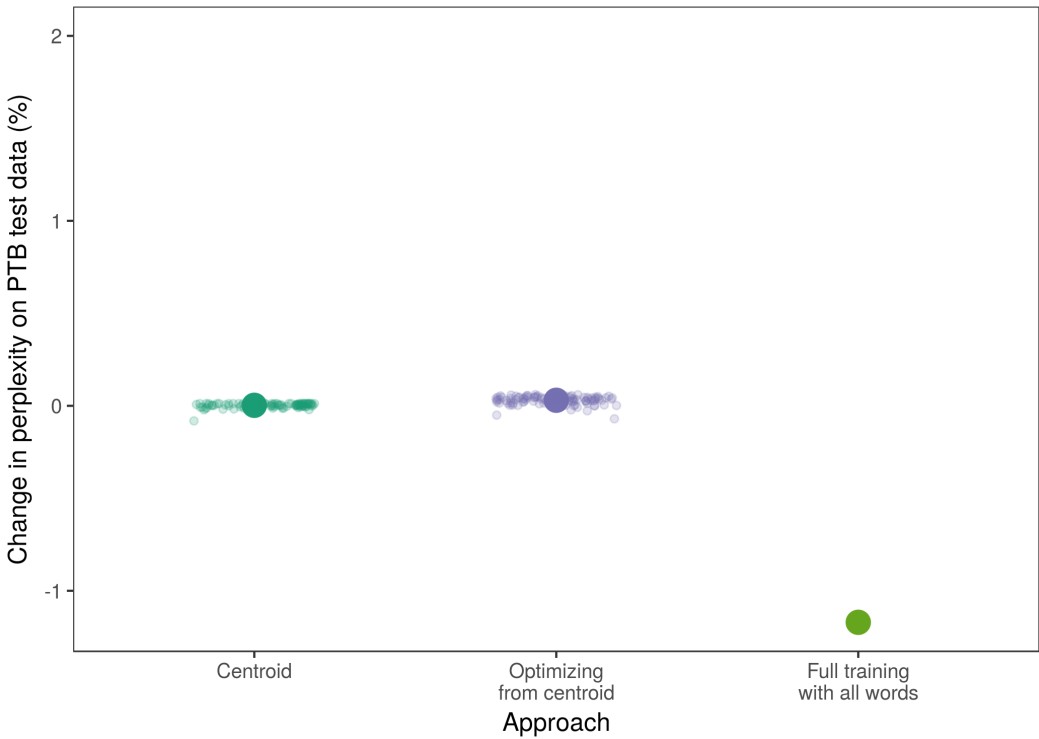

Figure 6: Percent change in perplexity on new word test data – comparing 10-shot learning across 100 different words.

## B    IMPLEMENTATION DETAILS

### B.1    MODEL

We used the "large" model described by Zaremba et al. (2014), and use all their hyper-parameters for the pre-training. Specifically, the model consists of 2 layers of stacked LSTMs with a hidden size of 1500 units, 35 recurrent steps, and dropout ($p_{keep} = 0.35$) applied to the non-recurrent connections. The gradients were clipped to a max global norm of 10. Weights were initialized uniformly from $[-0.04, 0.04]$.

#### B.1.1    TRAINING ON FULL CORPUS

The loss function was the cross-entropy loss of the model predictions. The model was trained for 55 epochs by stochastic gradient descent with a batch size of 20. The learning rate was set to 1 for the first 14 epochs, and then was decayed by a multiplier of $1/1.15$ per epoch for the remainder of training.

#### B.1.2    TRAINING ON A NEW WORD

**Centroid:** We extracted the input embeddings, softmax weights, and softmax biases of every word in the sentence except for the new word. We then averaged together these input embeddings to produce the new input embedding for the new word, etc.

**Optimizing:** We initialized the new input embedding in one of three ways: with the centroid from above, with a vector of zeros, or with the current embedding vector for the word (note that this last would not be possible in a general context where a new word is not "expected"). Starting from this point, we ran 100 epochs of batch gradient descent (batch size was equal to the number of training sentences), with a learning rate of 0.01. As a loss, we used the cross entropy loss plus 0.01 times the $\ell_2$ norm of the new embedding.

### B.1.3 LIST OF HUNDRED WORDS

```
['ab', 'absolutely', 'agricultural', 'aim', 'animals', 'announcing
 ↪ ', 'arguments', 'assist', 'averaged', 'bass', 'bullish', '
 ↪ calculations', 'carefully', 'claiming', 'compare', 'conceded
 ↪ ', 'congressman', 'consortium', 'contest', 'creation', '
 ↪ cumulative', 'danger', 'darman', 'die', 'discrimination', '
 ↪ disney', 'dominant', 'dorrance', 'edwards', 'efficiency', '
 ↪ elderly', 'enable', 'encouraging', 'entry', '
 ↪ environmentalists', 'execution', 'expenditures', 'facts', '
 ↪ formula', 'gaf', 'geneva', 'globe', 'golf', 'healthcare', '
 ↪ homeless', 'honor', 'horse', 'incest', 'informed', '
 ↪ investigators', 'iron', 'jackson', 'judgment', 'knight', '
 ↪ lake', 'lend', 'louisville', 'lowest', 'lucrative', '
 ↪ maturing', 'minute', 'mississippi', 'motorola', 'museum', '
 ↪ nabisco', 'netherlands', 'nigel', 'nine-month', 'owning', '
 ↪ petrochemical', 'pioneer', 'prepare', 'print', 'pro-choice',
 ↪  'recognized', 'referred', 'regarded', 'rejection', '
 ↪ requests', 'resorts', 'responsibilities', 'rolled', 'sansui'
 ↪ , 'serving', 'setback', 'similarly', 'somewhere', 'sounds',
 ↪ 'staffers', 'stolen', 'treasurys', 'treat', 'truth', 'utah',
 ↪  'vulnerable', 'ward', 'warsaw', 'wedtech', 'wheat', '
 ↪ wisconsin']
```

## C LOG-PROBABILITY ANALYSIS BROKEN OUT

| new_word | num_train | Approach | New word is correct | Wrong and irrelevant | Wrong but relevant |
|---|---|---|---|---|---|
| borrow | 1 | centroid | -9.75 | -10.43 | -9.65 |
| borrow | 1 | opt_centroid | -6.31 | -10.10 | -8.51 |
| borrow | 5 | centroid | -9.69 | -10.42 | -9.64 |
| borrow | 5 | opt_centroid | -5.78 | -10.44 | -8.71 |
| borrow | 10 | centroid | -9.65 | -10.42 | -9.61 |
| borrow | 10 | opt_centroid | -6.46 | -11.03 | -9.73 |
| borrow | 10 | with_word | -8.74 | -15.31 | -13.21 |
| cowboys | 1 | centroid | -9.33 | -10.40 | -9.06 |
| cowboys | 1 | opt_centroid | -7.57 | -8.88 | -8.84 |
| cowboys | 5 | centroid | -9.26 | -10.43 | -9.09 |
| cowboys | 5 | opt_centroid | -6.27 | -11.21 | -9.06 |
| cowboys | 10 | centroid | -9.26 | -10.42 | -9.10 |
| cowboys | 10 | opt_centroid | -6.23 | -11.10 | -9.71 |
| cowboys | 10 | with_word | -8.47 | -15.63 | -12.75 |
| immune | 1 | centroid | -9.15 | -10.53 | -9.63 |
| immune | 1 | opt_centroid | -8.99 | -9.17 | -9.39 |
| immune | 5 | centroid | -9.00 | -10.47 | -9.49 |
| immune | 5 | opt_centroid | -7.72 | -9.90 | -9.05 |
| immune | 10 | centroid | -9.01 | -10.50 | -9.49 |
| immune | 10 | opt_centroid | -6.23 | -11.03 | -8.96 |
| immune | 10 | with_word | -9.85 | -14.17 | -12.41 |
| rice | 1 | centroid | -8.67 | -10.44 | -9.64 |
| rice | 1 | opt_centroid | -7.65 | -10.09 | -8.80 |
| rice | 5 | centroid | -8.66 | -10.43 | -9.61 |
| rice | 5 | opt_centroid | -6.35 | -10.20 | -8.14 |
| rice | 10 | centroid | -8.66 | -10.42 | -9.61 |
| rice | 10 | opt_centroid | -5.85 | -10.47 | -8.64 |
| rice | 10 | with_word | -9.84 | -15.43 | -12.44 |

## D    Embedding similarity analyses

It has often been noted that relationships between word embeddings capture semantic features of the words, such as analogies between words (Mikolov et al., 2013). Thus it is natural to ask to what extent the word vectors produced by one-shot learning produce similarity structures close to those produced by full training with the word. To address this question, we computed the dot product[3] of the new word output embedding with all other word output embeddings. This vector of dot-product results can be thought of as a similarity map for the new word. We then compared these similarity maps to the similarity maps produced by full training with the word by computing correlations between the similarity maps. (See Fig. 7 for our results.)

Different runs of full training with the word produced very correlated similarity structures (all correlations greater than 0.9). Thus there does appear to be some consistent semantic structure that the embeddings are capturing. However, neither the centroid nor the optimizing one-shot learning methods seemed to be capturing this structure. The centroid method actually produced similarity structures that were **negatively** correlated with the full training similarity structures in three out of four cases! The optimizing method produced slightly more similar structures, but only in so far as most of its similarity structures were effectively uncorrelated with the full training similarity structure. Only for one word each did the methods produce a similarity structure that was strongly positively correlated with the structure produced by full training with the word.

These results are difficult to interpret. On the one hand, the consistency of the semantic knowledge produced by the network when fully trained with the word suggests that it is discovering important structure. On the other hand, the later learning approaches actually produce even more internally consistency, at least when training with several sentences. Full training with the word produces an average correlation of 0.93 between different training runs, whereas optimizing with all ten sentences produces an average correlation of 0.97 (and optimizing with 1 sentence produces an average correlation of 0.50). Furthermore, the results of section 3.2 above suggest that our approach is doing about as well at distinguishing contexts where the new word does and does not appear. That is, the optimizing runs are extracting consistent structure as well, they are just creating different representational structures than the full training runs.

These different representations may be explained by the removal of some contextual cues when the sentences are presented in isolation (noted above) – the networks doing full training with the word are able to pay attention to relationships across sentence boundaries, which may give them exposure to word co-occurrences that the one-shot learning approach is missing. Including the surrounding context sentences in the training might result in more similar representational structures.

---

[3]This seems to be the appropriate similarity "metric", given that the logits for the softmax over the vocabulary are formed by a matrix-vector multiplication which amounts to computing these dot products.

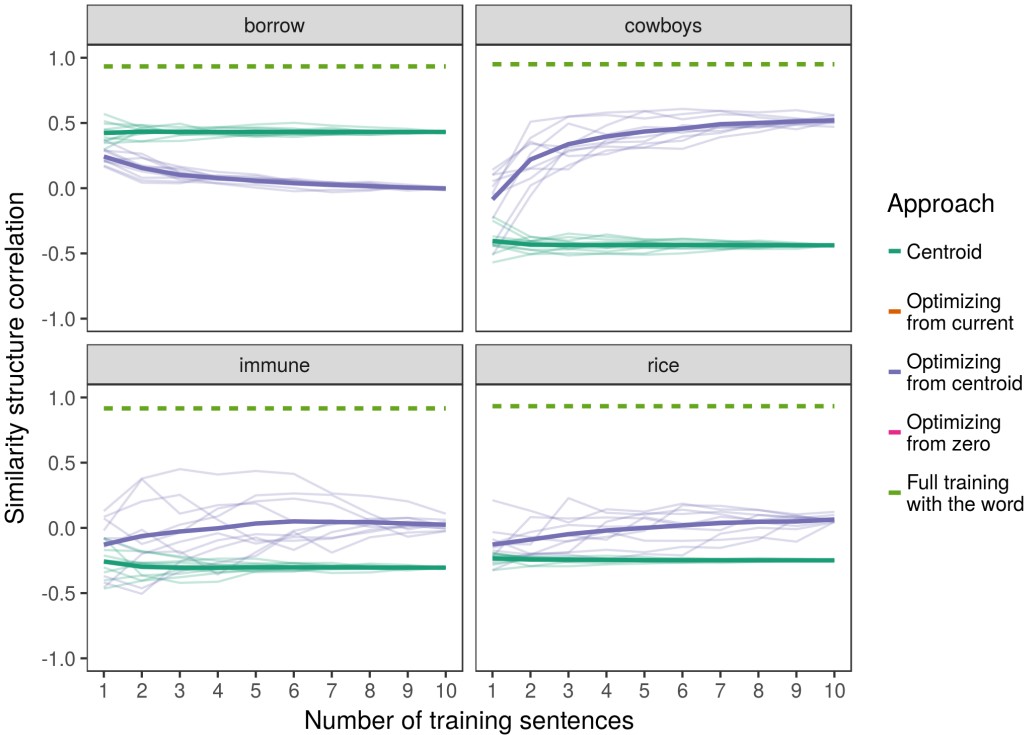

Figure 7: Similarity of representational similarity: how correlated are the similarity structures generated by the different methods with the similarity structure produced by training with the word?

# E   OTHER RELATED WORK

A commenter on a draft of this paper also noted that Herbelot & Baroni (2017) pursued related questions and independently came to some of the same conclusions we reached. However, we believe our results improve upon their work in several important ways. First, while they were only able to show benefits on training from definitions of a word, we show benefits of learning from a word in context. Furthermore, they only evaluated on embedding similarity, while we show behaviorally relevant improvements. This is important, because one conclusion from our results of our embedding similarity analyses in Appendix D show that dissimilar embeddings may nevertheless offer comparable performance. Finally, we explore in depth the effects of varying data parameters like the number of training sentences, and analyze where the improvements are occurring, both from the perspectives of parameters and outputs.

