# OpenReview forum: "One-shot and few-shot learning of word embeddings"
_ICLR.cc/2018/Conference — Reject_

### Official Review · AnonReviewer1 · 2017-11-25
**Worthwile goal of using backpropogation and weight-clamping to perform few-shot learning; questionable evaluation**

**Rating:** 4
**Confidence:** 4

**Review:**

I am highly sympathetic to the goals of this paper, and the authors do a good job of contrasting human learning with current deep learning systems, arguing that the lack of a mechanism for few-shot learning in such systems is a barrier to applying them in realistic scenarios. However, the main evaluation only considers four words - "bonuses", "explained", "marketers", "strategist" - with no explanation of how these words were chosen. Can I really draw any meaningful conclusions from such an experimental setup? Even the authors acknowledge, in footnote 1, that, for one of the tests, getting lower perplexity in three out of the four casess "may just be chance variation, of course". I wonder why we can't arrive at a similar conclusion for the other results in the paper. At the very least I need convincing that this is a reasonable experimental paradigm.

I don't understand the first method for initializing the word embeddings. How can we use the "current" embedding for a word if it's never been seen before? What does "current" mean in this context?

I also didn't understand the Latin square setup. Training on ten different permutations of the ten sentences suggests that all ten sentences are being used, so I don't see how this can lead to a few-shot or one-shot scenario.

---

> ### Author Response · Authors · 2018-01-02
> **Added an analysis of more words, and clarification of the smaller points raised.**
>
> All reviewers highlighted the small number of words we used. Briefly, we made this choice initially a) to allow us to explore the variation among different training sentences and different numbers of training sentences for the same word in more detail, and b) because our original experiments required training a new language model from scratch for each new word, which meant running many new words would require thousands of hours of compute time. However, it is useful to evaluate our approach across many words, and so we have added an experiment to the paper which does so while still maintaining computational feasibility.
>
> In this experiment, we selected 100 of the ~150 words that appear exactly 20 times in the PTB train corpus (omitting the words we used in prior experiments). Instead of training separate models without each word as we had previously, we trained a single model with NONE of these words. We then tested our one-shot learning technique and the centroid technique on these sentences, and compared to results obtained from ``full training with all words'' -- a model trained with the train sentences for each of the hundred words and the rest of the PTB training corpus. Notice that this comparison is not as precise as the earlier ones -- the ``full training with all words'' model receives about 2.5% more training data over all than any of the one-shot learning models, which means it will both perform better even on other words and will thus have more relevant linguistic structure to learn the new words from. Nevertheless, the comparisons between our technique and the centroid technique are still valid, and the comparison to the full training with all words gives a *worst-case* bound on how poorly one-shot methods will do compared to full training. In these experiments, we saw that our method performed both relatively consistently across different words, and performed consistently better than the centroid method. On average, it performed about as well as full training with the word (see the revised paper for full results).
>
> In order to partially compensate for the added material, we moved the embedding similairty analyses to the supplementary material, per reviewer 3's comment that they were unnecessary.
>
> Clarification of some minor details:
>
> Initializing from current embedding: You can imagine that a <new-word> token would be included in the softmax from the beginning, which could then be used as an initialization for any new words encountered. This would likely help the new word embeddings to be well separated from old embeddings (though it ultimately proves to be detrimental). Thanks for pointing out that this was not explained clearly, we have clarified this in the article as well.
>
> Latin square: We actually performed 100 training runs for each word, 10 runs corresponding to taking the first sentence from each of the 10 permutations, 10 runs corresponding to taking the first two sentences, etc. We've added a sentence to the paper that we hope will clarify this.

---

### Official Review · AnonReviewer2 · 2017-11-27

**Rating:** 3
**Confidence:** 4

**Review:**

The paper proposes a technique for exploiting prior knowledge to learn embedding representations for new words with minimal data. The authors provide a good motivation for the task and it is also a nice step in the general direction of learning deep nets and other systems with minimal supervision.

The problem is useful and very relevant to natural language applications, especially considering the widespread use of word embeddings within NLP systems. However, the demonstrated experimental results do not match the claims which seems a little grand. Overall, the empirical results is unsatisfactory. The authors pick a few example words and provide a detailed analysis. This is useful to understand how the test perplexity varies with #training examples for these individual settings. However, it is hardly enough to draw conclusion about the general applicability of the technique or effectiveness of the results. Why were these specific words chosen? If the reason is due to some statistical property (e.g., frequency) observed in the corpus, then why not generalize this idea and demonstrate empirical results for a class of words exhibiting the property. Such an analysis would be useful to understand the effectiveness of the overall approach. Another idea would be to use the one/few-shot learning to learn embeddings and evaluate their quality on a semantic task (as suggested in Section 3.3), but on a larger scale.

The technical contributions are also not novel. Coupled with the narrow experimentation protocol, it does not make the paper’s contributions or proposed claims convincing.

---

> ### Author Response · Authors · 2018-01-02
> **Added an analysis of more words to demonstrate the reliability of our approach**
>
> All reviewers highlighted the small number of words we used. Briefly, we made this choice initially a) to allow us to explore the variation among different training sentences and different numbers of training sentences for the same word in more detail, and b) because our original experiments required training a new language model from scratch for each new word, which meant running many new words would require thousands of hours of compute time. However, it is useful to evaluate our approach across many words, and so we have added an experiment to the paper which does so while still maintaining computational feasibility.
>
> In this experiment, we selected 100 of the ~150 words that appear exactly 20 times in the PTB train corpus (omitting the words we used in prior experiments). Instead of training separate models without each word as we had previously, we trained a single model with NONE of these words. We then tested our one-shot learning technique and the centroid technique on these sentences, and compared to results obtained from ``full training with all words'' -- a model trained with the train sentences for each of the hundred words and the rest of the PTB training corpus. Notice that this comparison is not as precise as the earlier ones -- the ``full training with all words'' model receives about 2.5% more training data over all than any of the one-shot learning models, which means it will both perform better even on other words and will thus have more relevant linguistic structure to learn the new words from. Nevertheless, the comparisons between our technique and the centroid technique are still valid, and the comparison to the full training with all words gives a *worst-case* bound on how poorly one-shot methods will do compared to full training. In these experiments, we saw that our method performed both relatively consistently across different words, and performed consistently better than the centroid method. On average, it performed about as well as full training with the word (see the revised paper for full results).
>
> In order to partially compensate for the added material, we moved the embedding similairty analyses to the supplementary material, per reviewer 3's comment that they were unnecessary.
>
> We agree that it would be exciting to see these methods applied to richer semantic tasks like the grounded tasks that we mentioned in the article, as several reviewers commented. However, it seems to us that our results are a useful starting place to demonstrate the method, and we are already straining the limits of the length of this paper.

---

### Official Review · AnonReviewer3 · 2017-11-28
**5: Marginally below the acceptance threshold.**

**Rating:** 4
**Confidence:** 4

**Review:**

Paper Summary

From just seeing a word used in a sentence, humans can infer a lot about this word by leveraging the surrounding words. Based on this idea, this work tries to obtain a better understanding of words in the one-shot or few-shot setting by leveraging surrounding word. They do this by language modeling sentences which contain rarely seen or never seen words. They evaluated their model using percent change in perplexity on test sentences containing new word by varying the number of training sentences containing this word. 3 Proposed Methods to model few-shot words: (1) beginning with random embedding, (2) beginning with zero embedding (3) beginning with the centroid of other words in the sentence. They compare to 2 Baseline Methods: (1) centroid of other words in the sentence, and (2) full training including the sparse words. Their results show that learning from centroids of other words can outperform full training on the new words.

Explanation
The paper is well written, and the experiments are well explained.  It is an interesting paper, and a research topic which is not well studied. The experiments are reasonable. The method seems to work well.

However, the method provides a very marginal difference between the previous method in Lazaridou et al. (2017). They just use backdrop to learn from this starting position. The main contribution of this work is the evaluation section.

Why only use the PTB language modeling task. Why not use the task in Gauthier & Mordatch or Hermann et al. The one task of language modeling shows promising results, but it’s not totally convincing.

One of the biggest caveats is that the experiments are only done in a few words. I’m not sure why more couldn’t have been done. This is discussed in section 4.1, but I think some of these differences could have been alleviated if there were more experiments done. Regardless, the experiments on the 8 words that they did chose were well done.

I don’t think that section 3.3 (embedding similarity) is particularly useful.

---

> ### Author Response · Authors · 2018-01-02
> **Added an analysis of more words to demonstrate the consistency of our technique.**
>
> All reviewers highlighted the small number of words we used. Briefly, we made this choice initially a) to allow us to explore the variation among different training sentences and different numbers of training sentences for the same word in more detail, and b) because our original experiments required training a new language model from scratch for each new word, which meant running many new words would require thousands of hours of compute time. However, it is useful to evaluate our approach across many words, and so we have added an experiment to the paper which does so while still maintaining computational feasibility.
>
> In this experiment, we selected 100 of the ~150 words that appear exactly 20 times in the PTB train corpus (omitting the words we used in prior experiments). Instead of training separate models without each word as we had previously, we trained a single model with NONE of these words. We then tested our one-shot learning technique and the centroid technique on these sentences, and compared to results obtained from ``full training with all words'' -- a model trained with the train sentences for each of the hundred words and the rest of the PTB training corpus. Notice that this comparison is not as precise as the earlier ones -- the ``full training with all words'' model receives about 2.5% more training data over all than any of the one-shot learning models, which means it will both perform better even on other words and will thus have more relevant linguistic structure to learn the new words from. Nevertheless, the comparisons between our technique and the centroid technique are still valid, and the comparison to the full training with all words gives a *worst-case* bound on how poorly one-shot methods will do compared to full training. In these experiments, we saw that our method performed both relatively consistently across different words, and performed consistently better than the centroid method. On average, it performed about as well as full training with the word (see the revised paper for full results), even though this is a worst-case bound. We think these results are quite encouraging, and hope they will address some of the concerns raised here.
>
> In order to partially compensate for the added material, we moved the embedding similairty analyses to the supplementary material, per your comment
>
> We agree that it would be exciting to see these methods applied to richer tasks like the grounded tasks that we mentioned in the article, as several reviewers commented. However, it seems to us that our results are a useful starting place to demonstrate the method, and we are already straining the limits of the length of this paper.

---

### Public Comment · ~Marco_Baroni1 · 2017-11-18
**Missing comparison?**

Nice paper, but motivation and methodology are very similar to the ones presented in:

Herbelot, A. and Baroni, M. 2017. High-risk learning: acquiring new word vectors from tiny data. In proceedings of the Conference on Empirical Methods in Natural Language Processing (EMNLP2017), Copenhagen, Denmark.

http://aclweb.org/anthology/D/D17/D17-1030.pdf

Perhaps, you could discuss how your proposal is different?

---

> ### Author Response · Authors · 2017-11-22
> **Thanks, and clarification of the contribution of our research**
>
> Thank you for the helpful reference! We had not encountered this work previously, and we agree that they share some similar features and will reference it in our final version of the paper. However, we think there are several features that distinguish our work:
>
> * First, their work only showed benefits of learning from definitional sentences, whereas ours demonstrates the ability to benefit from sentences which are not so clearly informative about the target word. This is important, because in practice when a sentence containing a new word is encountered, it is unlikely to conveniently be a definition of that word.
>
> * Furthermore, the only metric on which their approach shows improvement is the similarity of the produced embeddings to the "true" embedding. This may or may not be meaningful, since our representational similarity analyses suggest that there are dissimilar word embeddings that nevertheless produce similar performance in a complex task.
>
> * We demonstrate behaviorally relevant improvements (that is, our model's ability to do its task in the context of the new word improves).  We view this as an important part of exploring whether the representation learned is actually of real use in a language processing task.
>
> * Finally, we conducted more detailed analyses of the behavior and errors produced by our approach, such as the impact on prediction of other words and how this is affected by replay. We think these analyses provide important insights and caveats about our approach that will make it easier to refine and generalize.

---

> > ### Public Comment · ~Marco_Baroni1 · 2017-11-22
> > **Thanks for the clarification**
> >
> > Thanks for the clarification. I fully agree that, while the papers ask similar questions and propose similar approaches, yours is going further in various empirical ways.

---

### Decision · Program_Chairs · 2018-01-29
**ICLR 2018 Conference Acceptance Decision**

**Decision:**

Reject

**Comment:**

The paper is looking at an interesting problem, but it seems too early. The approach requires training a new language model  from scratch for each new word, rendering it completely impractical for real use. The main evaluation therefore only considers four words - "bonuses", "explained", "marketers", "strategist" (expanded to 20 during the rebuttal). This is not sufficient for ICLR.